# 53BP1 and USP28 mediate p53-dependent cell cycle arrest in response to centrosome loss and prolonged mitosis

Chii Shyang Fong[1], Gregory Mazo[1], Tuhin Das[1], Joshua Goodman[2], Minhee Kim[3], Brian P O'Rourke[1], Denisse Izquierdo[3], Meng-Fu Bryan Tsou[1,3]*

[1]Cell Biology Program, Memorial Sloan Kettering Cancer Center, New York, United States; [2]Oberlin College, Oberlin, United States; [3]BCMB Graduate Program, Weill Cornell Medical School, New York, United States

**Abstract** Mitosis occurs efficiently, but when it is disturbed or delayed, p53-dependent cell death or senescence is often triggered after mitotic exit. To characterize this process, we conducted CRISPR-mediated loss-of-function screens using a cell-based assay in which mitosis is consistently disturbed by centrosome loss. We identified 53BP1 and USP28 as essential components acting upstream of p53, evoking p21-dependent cell cycle arrest in response not only to centrosome loss, but also to other distinct defects causing prolonged mitosis. Intriguingly, 53BP1 mediates p53 activation independently of its DNA repair activity, but requiring its interacting protein USP28 that can directly deubiquitinate p53 in vitro and ectopically stabilize p53 in vivo. Moreover, 53BP1 can transduce prolonged mitosis to cell cycle arrest independently of the spindle assembly checkpoint (SAC), suggesting that while SAC protects mitotic accuracy by slowing down mitosis, 53BP1 and USP28 function in parallel to select against disturbed or delayed mitosis, promoting mitotic efficiency.

*For correspondence: tsoum@mskcc.org

## Introduction

Mitosis is a critical cell cycle phase during which duplicated chromosomes are correctly separated into two identical units on the spindle, restoring genome integrity after cell division. Various mechanisms exist to ensure proper mitosis with high accuracy. During mitotic entry at the G2/M border, cells can abort the process and return to G2 in response to damages or stresses via the antephase checkpoint (*Rieder and Cole, 1998*; *Matsusaka and Pines, 2004*; *Rudner and Murray, 1996*). After committing to mitotic entry, the accuracy of chromosome segregation is further protected by the spindle assembly checkpoint (SAC), which functions to delay anaphase onset until all chromosomes or kinetochores are properly attached to spindle microtubules (*Rieder and Cole, 2000*; *Lara-Gonzalez et al., 2012*). A wide range of cellular stresses are known to directly or indirectly disturb spindle assembly or chromosome segregation, including centrosome/kinetochore/microtubule dysfunctions (*Vitre and Cleveland, 2012*; *Bazzi and Anderson, 2014*), DNA damages (*Mikhailov et al., 2002*; *Carlson, 1950*; *Smits et al., 2000*; *Hut et al., 2003*), heat shock (*Maldonado-Codina et al., 1993*; *Vidair et al., 1993*; *Zajac et al., 2008*; *Erenpreisa et al., 2000*), hypoxia (*Fischer et al., 2004*; *Nystul et al., 2003*), and oxidative stress (*Kurata, 2000*), many of which activate SAC to slow down mitosis so that extra time is available for error correction. When mitotic stresses persist and errors are not fixed, cells can die via apoptosis during the prolonged arrest (*Topham, 2013*). In cases where cells go through mitosis in a normal time frame but divide in the presence of errors, cell cycle arrest can still be induced after mitotic exit. For example, a missegregating chromosome in anaphase is able to induce p53-dependent cell cycle arrest in the next G1

(*Hinchcliffe et al., 2016*). It has also been reported that missegregating chromosomes can be damaged by the cleavage furrow during cytokinesis and thereby activate an ATM-dependent DNA damage response in the following G1 (*Janssen et al., 2011*). In instances where missegregating chromosomes form micronuclei, they induce an ATR-dependent DNA damage response when the chromosomes undergo defective DNA replication that results in chromosome fragmentation and rearrangement (*Zhang et al., 2015*; *Crasta et al., 2012*). Missegregating chromosomes have also been shown to increase formation of protein aggregates in cells and induce a proteotoxic stress response (*Oromendia et al., 2012*). Moreover, some mitotic errors are known to cause cytokinesis failure, which in turn triggers p53 activation via the Hippo pathway (*Ganem et al., 2014*).

In addition to aforementioned damages, p53-dependent cell death or senescence has long been known as a stress response in cells going through division with a prolonged M phase (*Lanni and Jacks, 1998*; *Uetake and Sluder, 2010*). Depending on how long mitosis is disturbed or stalled, the trigger for p53 activation seems to vary, and in some cases, is perhaps a combination of many signals. A severe mitotic delay caused by prolonged treatments of spindle poisons can induce secondary damages such as telomere fusions, DNA damages, or leaky apoptosis, all of which contribute to p53 activation (*Hayashi et al., 2012*; *Orth et al., 2012*; *Colin et al., 2015*). Interestingly, however, even a mild stress in mitosis, such as centrosome loss (*Bazzi and Anderson, 2014*; *Lambrus et al., 2015*; *Izquierdo et al., 2014*; *Wong, 2015*; *Mikule et al., 2007*) or transient treatments of spindle poisons (*Uetake and Sluder, 2010*), which does not induce gross chromosome missegregation defects, cytokinesis errors, or other known secondary damages, can still efficiently trigger a widespread p53-dependent cell cycle arrest. Thus, regardless of what the actual trigger is in each case, it is clear that cells are sensitive to mitotic disturbance or delay, and that unfit cells undergoing erroneous or prolonged mitosis are selected against through various stress response mechanisms. To explore these selection processes in detail, here we used the centrosome loss as a model for mitotic stress, and performed a genome-wide CRISPR screen to identify essential molecules acting upstream or downstream of p53 in the pathway. We have surprisingly identified 53BP1 and USP28 as essential components acting upstream of p53 to mediate the stress response not only to centrosome loss, but also to other distinct defects that cause prolonged mitosis.

## Results

### A cell-based assay for centrosome loss-induced mitotic stress

Using diploid, non-transformed retinal pigment epithelial (RPE) cells, we constructed a stable $PLK4^{as}$ cell line in which the endogenous PLK4, a kinase specifically required for centrosome duplication (*Habedanck et al., 2005*; *Bettencourt-Dias et al., 2005*), was replaced with an analog-sensitive mutant (PLK4$^{as}$) that could be chemically inactivated by the ATP analog 3MBPP1 (see Materials and methods) (*Kim, 2016*). Upon PLK4 inactivation, cells were gradually depleted of centrosomes (*Figure 1—figure supplement 1*), and started to divide more slowly with mitotic duration increasing to ~100 min instead of ~30 min observed in control cells (*Figure 1A*). Within a few days, all acentrosomal cells stopped proliferating (*Figure 1B*), and were arrested in G1 with high levels of nuclear p53 and p21 (*Figure 1C and D*), consistent with a previous report (*Wong, 2015*). Removal of p53 (*Figure 1—figure supplement 2*), however, alleviated both the growth arrest (*Figure 1E*) and nuclear accumulation of p21 (*Figure 1F*), but not mitotic delay (*Figure 1G*), allowing acentrosomal cells to continue proliferating in the presence of mitotic stress at rates not significantly different from control or unstressed cells (*Figure 1E*). We thus established a genetically defined, chemically inducible assay in which the p53-dependent G1 arrest induced by centrosome loss could be uniformly activated and thus systematically dissected.

### CRISPR-mediated, loss-of-function screens for components acting upstream or downstream of p53 in response to centrosome loss

Using this system, we carried out a genome-wide CRISPR-mediated loss-of-function screen for genes whose inactivation enabled $PLK4^{as}$; $p53^{+/+}$ cells to survive and proliferate in the absence of centrosomes (*Figure 2A*). Eight independent screens were performed using a pooled lentivirus sgRNA library covering >95% of human genes (*Sanjana et al., 2014*; *Shalem et al., 2014*), with each gene targeted by at least 6 different sgRNAs. sgRNAs carried or enriched by survivors were analyzed by

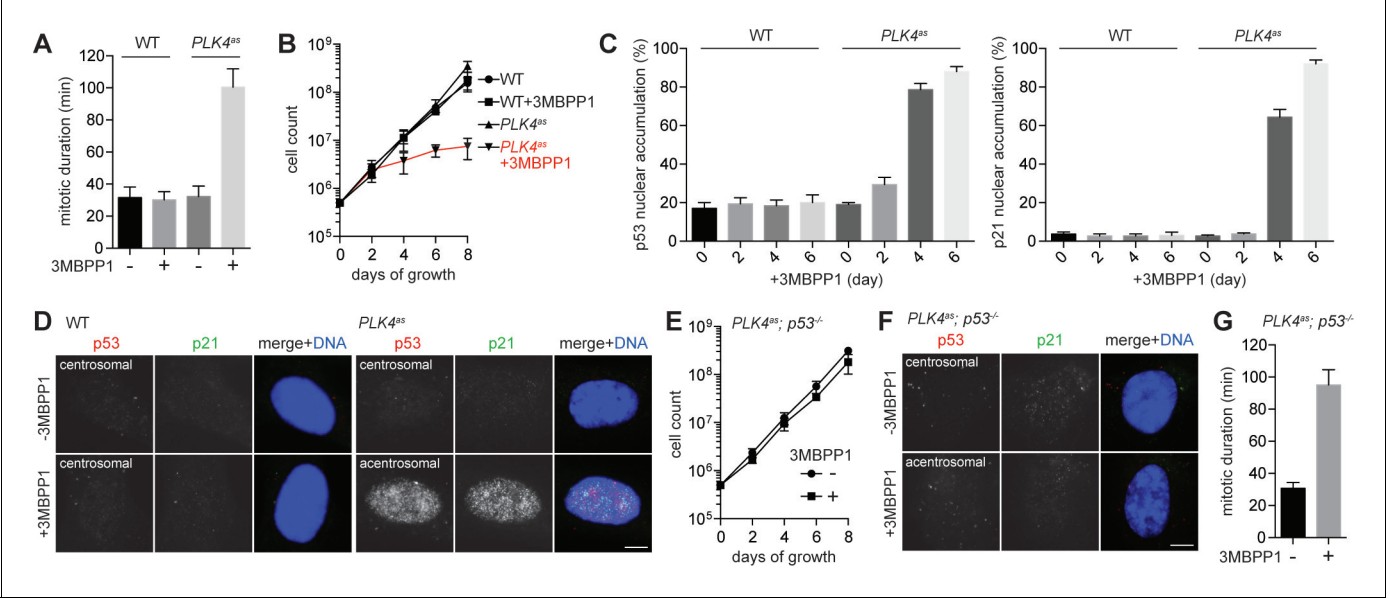

**Figure 1.** Genome-wide CRISPR-mediated loss-of-function screen for components required for centrosome loss-induced G1 arrest. (A) Acentrosomal cells exhibits prolonged mitosis. Measurement of mitotic duration of wild type RPE1 and *PLK4^as* cells dividing in the presence or absence of 3MBPP1 with live-cell imaging. With 3MBPP1 treatment, cells gradually lost centrosomes and ceased to proliferate; the duration of acentrosomal mitosis was measured four days after 3MBPP1 addition. Data are means ± SD. *n*>30, *N* = 3. (B) Cell proliferation ceases in acentrosomal cells. Growth curve of wild type RPE1 and *PLK4^as* cells with or without 3MBPP1 treatment. Data are means ± SD. *n*>50, *N* = 3. (C) p53 and p21 accumulate in the nucleus of acentrosomal cells. Quantification of p53 (left) and p21 (right) nuclear accumulation in wild type RPE1 and *PLK4^as* cells after 3MBPP1 addition. Data are means ± SD. *n*>100, *N* = 3. (D) Representative immunofluorescence images of cells in (C) on day 6 stained with antibodies against p53 and p21. Scale bar, 5 μm. (E) Acentrosomal cells continue to proliferate when p53 is removed. The growth curve of *PLK4^as*; *p53^-/-* cells following 3MBPP1 addition. Refer to (B) for growth curves of *PLK4^as*cells. Data are means ± SD. *n*>50, *N* = 3. (F) p21 does not accumulate in *PLK4^as*; *p53^-/-* cells during acentrosomal cell division. Immunofluorescence images of cells stained with the antibodies indicated. Scale bar, 5 μm. (G) *PLK4^as*; *p53^-/-* cells divide by prolonged mitosis in the absence of the centrosome. Graph showing mitotic duration of centrosomal and acentrosomal *PLK4^as*; *p53^-/-* cells measured with live-cell imaging. Data are means ± SD. *n*>30, *N* = 3.

The following figure supplements are available for figure 1:

**Figure supplement 1.** Centrosome loss upon PLK4 inactivation.

**Figure supplement 2.** Genotyping of p53 CRISPR cell line.

deep sequencing to reveal the targeted genes, and 27 candidate genes were identified (*Figure 2B* and *Table 1*). sgRNAs for 5 genes were most highly enriched (*Figure 2B* and *Table 1*), including the previously known p53 and p21, and three novel genes, 53BP1, USP28, and TRIM37 that have not been linked to centrosome loss-induced G1 arrest. Moreover, for these 5 genes, at least 3 out of the 6 sgRNAs were repeatedly enriched in independent screens (*Table 1*), suggesting that they are unlikely false positive hits. 53BP1 is a known key player in DNA double-strand break (DSB) repair (*Panier and Boulton, 2014*), but was first characterized as a binding partner of p53, albeit with unclear functions (*Thukral et al., 1994*; *Iwabuchi et al., 1994*). USP28 is a deubiquitinating enzyme known to interact with 53BP1 (*Zhang et al., 2006*), but it puzzlingly has minor or no role in DSB repair (*Knobel et al., 2014*), raising an interesting possibility that perhaps 53BP1 and USP28 have a specific role in centrosome loss-induced G1 arrest. Apart from 53BP1, no other sgRNAs targeting major DNA damage response (DDR) components such as ATM, MDC1, RNF8 or BRCA1 were enriched in our screen (*Figure 2B*), even though they could be repeatedly detected in the baseline reads in all independent screens. To ensure the specificity of the results, we verified these top hits by creating individual CRISPR cell lines in the *PLK4^as* background (see Materials and methods; *Figure 2—figure supplement 1*) and assessed their growth in the presence or absence of centrosomes. Similar to *p53^-/-* cells, clonal *53BP1^-/-*, *USP28^-/-*, *TRIM37^-/-*, and *p21^-/-* cell lines continued to

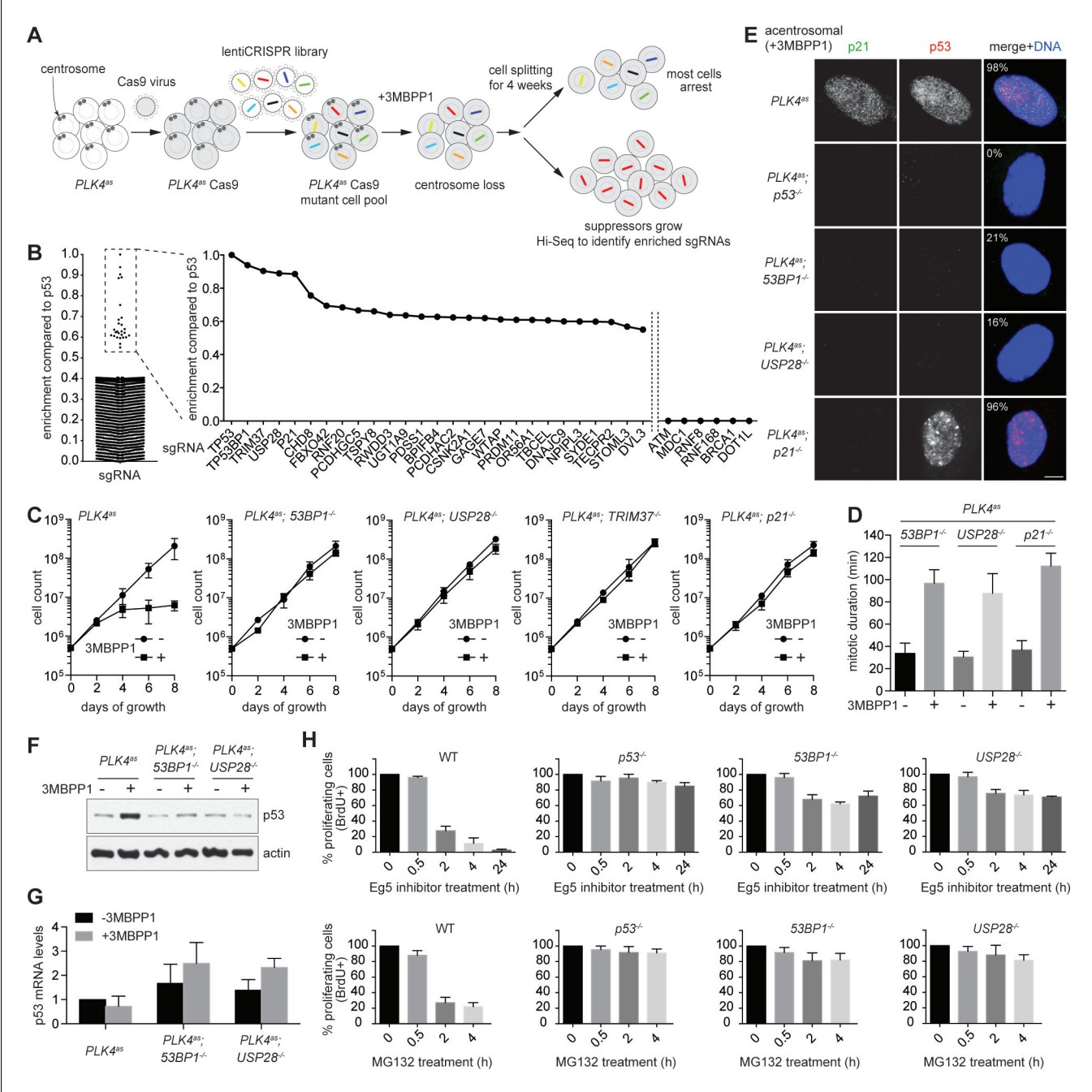

**Figure 2.** 53BP1 and USP28 are broad acting components acting upstream of p53 in response to mitotic stress. (**A**) Schematic representation of the loss-of-function screen for components required for centrosome loss induced G1 arrest using the human genome-scale CRISPR knockout library (GeCKO). (**B**) Binary logarithm of sgRNA enrichment for genes potentially involving in centrosome loss induced G1 arrest, normalized to the top scoring hit p53. The HiSeq data was collected from eight independent screens. A candidate gene must be hit repeatedly with high scores (HiSeq reads) by two or more of its six sgRNAs in independent screens. Interesting negative hits were also shown, including critical DDR components whose sgRNAs were not enriched but detected only in the baseline reads. (**C**) Validation of top five scoring hits from the screen other than the p53 control. The results shown here used clonal 53BP1$^{-/-}$, USP28$^{-/-}$, and p21$^{-/-}$ CRISPR knockout cell lines derived from PLK4$^{as}$ knock-in cells (*PLK4$^{as}$-KI*) obtained from A. Holland (***Moyer et al., 2015***) (see Materials and methods). The growth curve of the indicated individual CRISPR cell lines in the presence or absence of 3MBPP1 was shown. Data are means ± SD. *n*>50, *N* = 3. (**D**) Acentrosomal *53BP1$^{-/-}$, USP28$^{-/-}$* and *p21$^{-/-}$* cells proliferate in the presence of mitotic delay. Graph showing mitotic duration of the indicated CRISPR cell lines dividing with or without centrosomes measured with live-cell imaging. Data are means ± SD. *n*>30, *N* = 3. (**E**) 53BP1 and USP28 function upstream of p53 to activate G1 arrest. Immunofluorescence images of CRISPR cell lines grown in 3MBPP1 stained with the indicated antibodies. The percentage in the merged panel indicates the proportion of cells with p53 nuclear accumulation. Scale bar, 5 μm. (**F**) Total p53 levels are not elevated in acentrosomal *53BP1$^{-/-}$, USP28$^{-/-}$* cells. Immunoblot for p53 protein of the indicated cell lines grown in the

*Figure 2 continued on next page*

*Figure 2 continued*

presence or absence of 3MBPP1 for seven days. (G) p53 protein elevation during centrosome loss-induced G1 arrest is not due to increased p53 transcription. Quantification of p53 mRNA levels relative to GAPDH in (F) by qRT-PCR. Data are means ± SD. *n* = 6 from two independent experiments. (H) *53BP1⁻/⁻*, *USP28⁻/⁻* cells do not arrest in G1 despite experiencing mitotic stresses induced by different drug treatments. BrdU incorporation assay for 24 hr showing proportion of proliferating cells in the indicated CRISRP cell lines following release into mitosis with different duration of Eg5 inhibitor (top panels) and MG132 (bottom panels) treatment and washout. Percentages are normalized to the untreated control. Data are means ± SD. *n*>250, *N* = 3.

The following figure supplements are available for figure 2:

**Figure supplement 1.** Genotyping of CRISPR cell lines.

**Figure supplement 2.** Schematic outlining the timeline of synchronization and drug treatments used.

proliferate regardless of whether the centrosomes are present or not (*Figure 2C*), validating our screen. Analyses of TRIM37, however, indicate that it is involved in a distinct cellular process (not shown), and thus will be addressed elsewhere. Here we focus our report on 53BP1 and USP28, and their relationships with p53 and p21.

Similar to $p53^{-/-}$ cells, $53BP1^{-/-}$, $USP28^{-/-}$ and $p21^{-/-}$ cells were found to actively proliferate with normal mitotic duration in the presence of centrosomes (*Figure 2D*), indicating that these genes do not have a direct role in regulating mitotic progression. In the absence of centrosomes, however, $53BP1^{-/-}$, $USP28^{-/-}$ and $p21^{-/-}$ cells experienced a similar 90-minute mitotic delay (*Figure 2D*), an indication of mitotic disturbance or stress, but could not activate cell cycle arrest (*Figure 2C*), suggesting that 53BP1, USP28 and p21, together with p53, have an important role in centrosome loss-induced G1 arrest. To test whether 53BP1 and USP28 act downstream or upstream of p53 in the pathway, we examined p53 accumulation in cells. Upon centrosome removal, nuclear p53 was detected in nearly all $PLK4^{as}$ cells arrested in G1 (*Figure 1C*, left). The same result was seen for acentrosomal $p21^{-/-}$ cells (*Figure 2E*) albeit actively proliferating, consistent with p21 acting downstream of p53 to execute G1 arrest (*Wong, 2015*; *Stewart et al., 1999*). Conversely, the frequency of nuclear p53 accumulation in acentrosomal $53BP1^{-/-}$ or $USP28^{-/-}$ cells (*Figure 2E*) was no different from that of unstressed, centrosomal cells (*Figure 1C*, left). Similarly, western blot analyses revealed that the total p53 levels in $53BP1^{-/-}$ or $USP28^{-/-}$ cells were kept low during acentrosomal cell division in the presence of mitotic delay (*Figure 2F*), indicating that 53BP1 and USP28 function upstream of p53 to initiate cell cycle arrest in response to centrosome loss. Note that the elevation in p53 protein levels during centrosome loss-induced G1 arrest was likely due to a post-transcription event, as we observed no increase in p53 mRNA levels during the arrest (*Figure 2G*). We have thus established the first vertebrate cell lines, $53BP1^{-/-}$ or $USP28^{-/-}$ cells, in which centrosomes can be stably lost without abolishing the entire p53 network.

## 53BP1 and USP28 are required for G1 arrest induced by different mitotic stresses

We next asked whether 53BP1 and USP28 are part of the G1 arrest machinery specific to the loss of centrosome or more general to other cellular damages that induce mitotic stress/delay. We tested the ability of $53BP1^{-/-}$ and $USP28^{-/-}$ cells to proliferate after experiencing mitotic delay caused by an Eg5 inhibitior (dimethylenastron) or proteasome inhibitor (MG132) treatment. Eg5 inhibitor and MG132 induce mitotic delay through distinct mechanisms, with the former a spindle poison disrupting spindle bipolarity, and the latter blocking cyclin B destruction without affecting spindle assembly. To this end, cells were synchronized at G2-M transition by CDK1 inhibitor RO-3306, after which they were released into mitosis in the presence of Eg5 inhibitor or MG132. Mitotic cells were then shaken off at defined time points, washed of the drugs and cultured for 24 hr in the presence of bromodeoxyuridine (BrdU) (*Figure 2—figure supplement 2*). While most wild-type cells exposed to Eg5 inhibitor or MG132 for 2 hr or more did not show BrdU incorporation, an indication of a cell cycle arrest in G1, $p53^{-/-}$ cells continued to proliferate under the same conditions (*Figure 2H*). Strikingly, similar to $p53^{-/-}$ cells, most $53BP1^{-/-}$ and $USP28^{-/-}$ cells also continued to progress through the cell

**Table 1.** Candidate genes enriched in the eight independent screens for genes involved in centrosome loss-induced G1 arrest.

| Enrichment compared to p53 | Gene symbol | Gene name or description | Number of sgRNAs enriched out of total 6 |
|---|---|---|---|
| 1.0000 | TP53 | Tumor Protein P53 | 5/6 |
| 0.9386 | TP53BP1 | Tumor Protein P53 Binding Protein 1 | 4/6 |
| 0.9039 | TRIM37 | Tripartite Motif Containing 37 | 3/6 |
| 0.8895 | USP28 | Ubiquitin Specific Peptidase 28 | 4/6 |
| 0.8858 | P21 | Cyclin-Dependent Kinase Inhibitor 1A (P21, Cip1) | 6/6 |
| 0.7559 | CHD8 | Chromodomain Helicase DNA Binding Protein 8 | 4/6 |
| 0.6945 | FBXO42 | F-Box Protein 42 | 3/6 |
| 0.6846 | RNF20 | Ring Finger Protein 20, E3 Ubiquitin Protein Ligase | 3/6 |
| 0.6665 | PCDHGC5 | Protocadherin Gamma Subfamily C, 5 | 2/6 |
| 0.6604 | TSPY8 | Testis Specific Protein, Y-Linked 8 | 3/6 |
| 0.6395 | RWDD3 | RWD Domain Containing 3 | 2/6 |
| 0.6364 | UGT1A9 | UDP Glucuronosyltransferase 1 Family, Polypeptide A9 | 2/6 |
| 0.6285 | PDSS1 | Prenyl (Decaprenyl) Diphosphate Synthase, Subunit 1 | 2/6 |
| 0.6275 | BPIFB4 | BPI Fold Containing Family B, Member 4 | 2/6 |
| 0.6240 | PCDHAC2 | Protocadherin Alpha Subfamily C, 2 | 3/6 |
| 0.6218 | CSNK2A1 | Casein Kinase 2, Alpha 1 Polypeptide | 2/6 |
| 0.6201 | GAGE7 | G Antigen 7 | 3/6 |
| 0.6119 | WTAP | Wilms Tumor 1 Associated Protein | 2/6 |
| 0.6093 | PRDM11 | PR Domain Containing 11 | 2/6 |
| 0.6086 | OR56A1 | Olfactory Receptor, Family 56, Subfamily A, Member 1 | 2/6 |
| 0.6059 | TBCEL | Tubulin Folding Cofactor E-Like | 2/6 |
| 0.5997 | DNAJC9 | DnaJ (Hsp40) Homolog, Subfamily C, Member 9 | 2/6 |
| 0.5991 | NPIPL3 | Nuclear Pore Complex Interacting Protein Family, Member B3 | 3/6 |
| 0.5987 | SYDE1 | Synapse Defective 1, Rho GTPase, Homolog 1 | 2/6 |
| 0.5959 | TECPR2 | Tectonin Beta-Propeller Repeat Containing 2 | 2/6 |
| 0.5683 | STOML3 | Stomatin (EPB72)-Like 3 | 2/6 |
| 0.5507 | DVL3 | Dishevelled Segment Polarity Protein 3 | 2/6 |

Log$_2$ scaling of sgRNAs enrichment normalized to the top scoring hit p53. The sgRNAs for genes highlighted in red were validated in this study. For a gene to be scored as a candidate, at least two of its six sgRNAs were repeatedly enriched in independent screens, with the HiSeq reads at least 3 times higher than the average of the baseline read.

cycle after exposure to the drugs (**Figure 2H**), demonstrating that 53BP1 and USP28 are broadly required for G1 arrest induced by different mitotic stresses.

## 53BP1 mediates centrosome loss-induced G1 arrest independent of its DNA repair activity

It was intriguing that a critical component of DDR was identified in our screen after several reports had suggested that DDR is not involved in the G1 arrest induced by acentrosomal division (**Bazzi and Anderson, 2014**; **Wong, 2015**). This led us to hypothesize that 53BP1 mediates the G1 arrest independently of its DDR role. Inspection of 53BP1 sequence revealed that apart from the multiple domains and motifs that are critical for DDR function (**Panier and Boulton, 2014)**, the C--terminal tandem BRCT domain is not (**Ward et al., 2006**) (**Figure 3A**). Interestingly, the tandem BRCT domain is known to interact with p53 and USP28 (**Iwabuchi et al., 1994**; **Knobel et al., 2014**; **Joo et al., 2002**; **Derbyshire et al., 2002**). We hence speculated that the tandem BRCT domain is

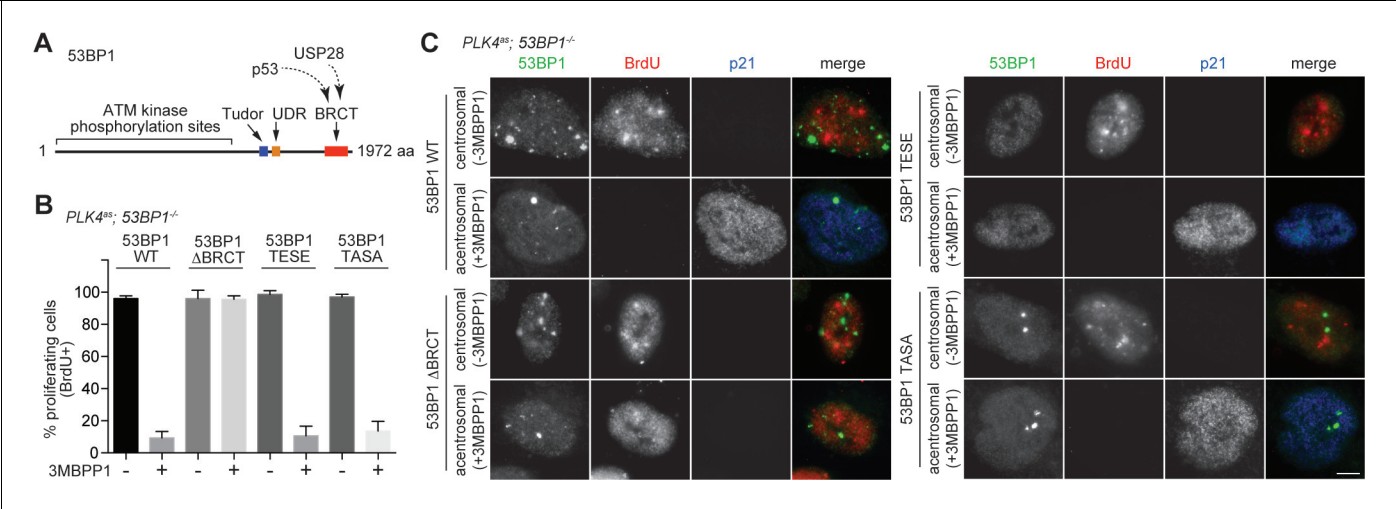

**Figure 3.** 53BP1 mediates centrosome loss-induced G1 arrest independently to its DNA repair activity. (A) Domain organization of 53BP1. BRCT (BRCA1 carboxy-terminal), UDR (ubiquitylation-dependent recruitment). p53 and USP28 interact with 53BP1 through the tandem BRCT domain. (B) 53BP1$^{\Delta BRCT}$ mutant does not rescue the G1 arrest in $PLK4^{as}$; $53BP1^{-/-}$ cells after centrosome removal. Wild-type or indicated mutant 53BP1 were mildly expressed under the tetracycline inducible promoter in stable, clonal, centrosomal $PLK4^{as}$; $53BP1^{-/-}$ cells (see Materials and methods), during which centrosome loss was induced by 3MBPP1 addition. BrdU was added on day six after 3MBPP1 addition and cells were harvested 24 hr later for BrdU incorporation assay (3MBPP1 treatment for seven days in total). Data are means ± SD. $n>150$, $N = 3$. (C) Representative immunofluorescence images of cells in (B) stained with the indicated antibodies seven days after 3MBPP1 treatment. 53BP1 was stained with anti-GFP FITC conjugated antibody. Scale bar, 5 μm.

The following figure supplements are available for figure 3:

**Figure supplement 1.** Wild type and mutant 53BP1 are exogenously expressed to similar levels in $PLK4^{as}$; $53BP1^{-/-}$ cells.

**Figure supplement 2.** Schematic outlining the timeline of drug treatments used.

**Figure supplement 3.** DDR function is intact in 53BP1$^{WT}$, 53BP1$^{\Delta BRCT}$ and 53BP1$^{TASA}$, but not in 53BP1$^{TESE}$.

required for the G1 arrest induced by centrosome loss. To test this, we stably expressed exogenous wild type 53BP1 or 53BP1 BRCT-deletion mutant (53BP1$^{\Delta BRCT}$) in $PLK4^{as}$; $53BP1^{-/-}$ cells (**Figure 3— figure supplement 1**), inactivated PLK4 to induce centrosome loss, and assessed whether the G1 arrest could be rescued (**Figure 3—figure supplement 2**). While ~90% of the cells expressing wild-type 53BP1 arrested in G1 upon centrosome removal, reintroduction of the 53BP1$^{\Delta BRCT}$ mutant failed to do so, allowing acentrosomal cells to proliferate in the presence of mitotic stress/delay (**Figure 3B and C**). DDR response in cells expressing the 53BP1$^{\Delta BRCT}$ was intact as the mutant protein localized to DNA damage sites marked by γ-H2AX (**Figure 3—figure supplement 3**), consistent with previous reports (**Ward et al., 2006**). Our results thus demonstrate that centrosome loss-induced G1 arrest requires the DDR-independent tandem BRCT domain of 53BP1.

Next, we sought to determine if the DDR activity of 53BP1 is required for centrosome loss-induced G1 arrest. 53BP1 is phosphorylated by mitotic kinases at T1609 and S1618 within the UDR motif to inhibit its recruitment to DSB sites during mitosis, an important regulatory mechanism preventing deleterious telomeric fusion that otherwise can occur during prolonged mitosis (**Hayashi et al., 2012**; **Orthwein et al., 2014**). Capitalizing on this inhibitory mechanism, we asked whether reintroduction of a phosphomimetic mutant of 53BP1 (53BP1$^{TESE}$) could rescue the G1 arrest of $PLK4^{as}$; $53BP1^{-/-}$ cells dividing in the absence of the centrosome. We found that despite lacking DDR function as indicated by its delocalization from DNA damage sites (**Figure 3—figure supplement 3**), the 53BP1$^{TESE}$ mutant could robustly rescue G1 arrest in response to mitotic stress (**Figure 3B and C**), indicating that the DDR function of 53BP1 is not required for the G1 arrest. To further determine if the mitotis-specific phosphorylation of 53BP1, which inactivates its DDR function, is required for 53BP1 to mediate G1 arrest, we expressed the constitutively active, phosphonull

form of 53BP1 (53BP1$^{TASA}$) in *PLK4$^{as}$; 53BP1$^{-/-}$* cells. Intriguingly, 53BP1$^{TASA}$ efficiently rescued the arrest in G1 upon centrosome loss (*Figure 3B,C* and *Figure 3—figure supplement 3*). Together, our results indicate that 53BP1 can efficiently mediate the centrosome loss-induced G1 arrest regardless of whether it is active for DDR.

## USP28 mediates centrosome loss-induced G1 arrest through its deubiquitinase activity, and acts downstream of 53BP1 to stabilize p53

USP28 is a deubiquitinating enzyme carrying two conserved catalytic domains – UCH-1 (Cys box) and UCH-2 (His box) (*Figure 4A*). To test whether USP28 mediates centrosome loss-induced G1 arrest through its enzymatic activity, we mutated the catalytic cysteine (C171) and histidine (H600) to alanine and examined the consequence on the G1 arrest. We found that unlike wild-type USP28, expression of the catalytic-inactive USP28 (USP28$^{CI}$) in *PLK4as; USP28$^{-/-}$* cells (*Figure 4—figure supplements 1* and *2*) failed to rescue the G1 arrest induced by centrosome loss, phenocopying the loss of USP28 (*Figure 4B,C*), indicating that the catalytic activity of USP28 is essential for the G1 arrest. Moreover, purified USP28 was found to directly deubiquitinate p53 in vitro (*Figure 4D*), raising potentially a direct role of USP28 in stabilizing p53 in vivo. Consistently, overexpression of the wild type USP28 but not USP28$^{CI}$ in normal, unstressed cells caused ectopic nuclear p53 accumulation and cell cycle arrest uniformly across the entire population (100%, *Figure 4E*; not shown). The nuclear p53 accumulation caused by overexpression of wild type USP28 was not due to a specific increase in p53 mRNA levels (*Figure 4F*), further supporting our observation that USP28 deubiquitinates p53 for protein stabilization. To determine the relationship between 53BP1 and USP28, we examined whether the ectopic stabilization of p53 by USP28 depends on 53BP1, and vice versa. We found that upon USP28 overexpression, p53 was robustly stabilized even in *53BP1$^{-/-}$* cells (*Figure 4G*), indicating that USP28 does not act upstream of 53BP1 to control p53 level. Intriguingly, similar to USP28, overexpression of wild type 53BP1 but not 53BP1$^{\Delta BRCT}$ also efficiently induced ectopic p53 stabilization in unstressed cells (*Figure 4H*). However, unlike USP28 that can act without 53BP1, overexpression of 53BP1 had no effect on p53 level in *USP28$^{-/-}$* cells (*Figure 4I*). Thus, consistent with the known interaction of 53BP1 with USP28, we show that while 53BP1 has no deubiquitinase activity, it can function upstream of USP28 to stabilize p53.

## 53BP1 disassociates from kinetochores irreversibly in a time-dependent manner, and forms nuclear foci with p53 and USP28 in response to mitotic stress

To explore how 53BP1 or USP28 might respond to mitotic stress, we examined their localizations in mitosis. In normal cells, 53BP1, but not USP28, was found to localize to kinetochores during early mitosis when spindle assembly checkpoint (SAC) was on (*Figure 5A* left, bright BubR1 signal), and delocalize at anaphase when SAC was off (*Figure 5A* left, weak BubR1), a pattern resembling that of SAC components as documented previously (*Jullien et al., 2002*). We next examined if 53BP1 behaves like SAC components during mitotic stress. Upon centrosome loss (*Figure 5A* right) or Eg5 inhibition (*Figure 5—figure supplement 1A*), cells were delayed in prometaphase during which SAC was on, as indicated by bright BubR1 (*Figure 5A* right; *Figure 5—figure supplement 1A*) or Mad2 (*Figure 5—figure supplement 1B*) at kinetochores. Strikingly, under the same conditions, 53BP1 was often seen to be absent from the kinetochore, suggesting that 53BP1 is not a typical SAC component. To test whether 53BP1 disassociates from kinetochore in a time-dependent manner, cells were synchronized at G2/M, released into stressed mitosis with activated SAC, and examined for kinetochore 53BP1 (*Figure 5—figure supplement 2A*). Intriguingly, in cells stressed with centrosome loss (*Figure 5B* left) or Eg5 inhibition (*Figure 5B* right), 53BP1 gradually disassociated from kinetochores, starting from ~30 min after mitotic entry, whereas the SAC components BubR1 or Mad2 were consistently detected at kinetochores during the lengthened prometaphase. These results suggest that the kinetochore localization of 53BP1 is time-sensitive, rather than dependent on SAC. To further examine the relationship between kinetochore 53BP1 and SAC, we asked whether reactivation of SAC after it was turned off could relocalize 53BP1 to mitotic kinetochores. We inactivated SAC by arresting cells in metaphase with MG132 treatment, as indicated by the weak BubR1 or the absence of Mad2 signals at the kinetochores, and then reactivated SAC with a transient pulse of nocodazole treatment (*Figure 5—figure supplement 2B*). We found that

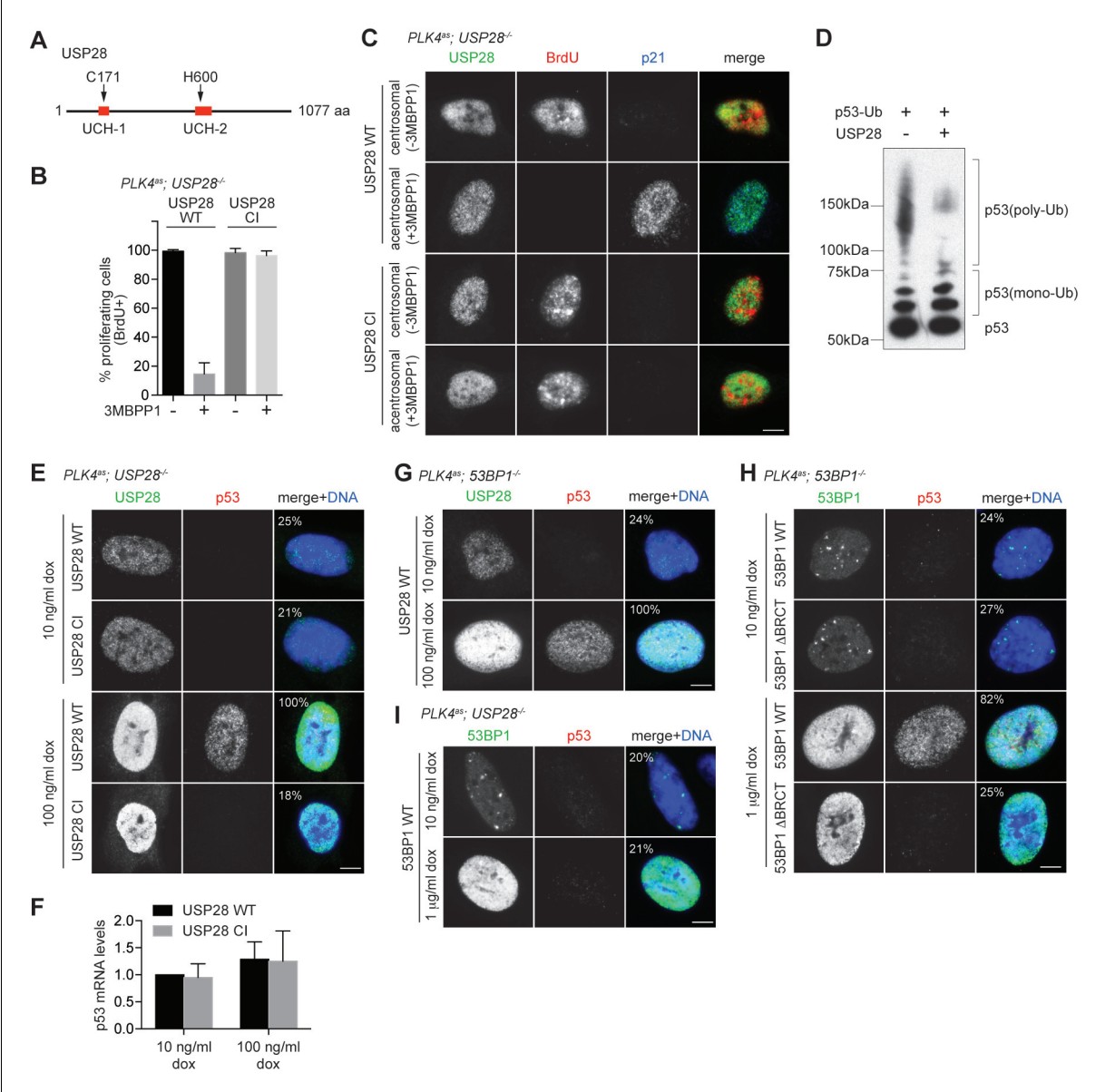

**Figure 4.** USP28 is catalytically required for p53 stabilization during centrosome loss-induced G1 arrest. (**A**) Organization of the conserved catalytic domains in USP28. UCH-1 (Cys box, amino acids 162–196) and UCH-2 (His box, amino acids 580–649). (**B**) Catalytic-inactive USP28$^{CI}$ cannot rescue the G1 arrest in *PLK4$^{as}$; USP28$^{-/-}$* cells after centrosome removal. Wild type UPS8 or USP28$^{CI}$ was mildly expressed under the tetracycline inducible promoter in stable, clonal, centrosomal *PLK4$^{as}$; USP28$^{-/-}$* cells (see Materials and methods) during which centrosome loss was induced by 3MBPP1 addition. BrdU was added on day six after 3MBPP1 addition and cells were harvested 24 hr later for BrdU incorporation assay (3MBPP1 treatment for seven days in total). Data are means ± SD. *n*>150, *N* = 3. (**C**) Representative immunofluorescence images of cells in (**B**) stained with the indicated antibodies seven days after 3MBPP1 treatment. USP28 was stained with anti-HA antibody. Scale bar, 5 μm. (**D**) USP28 deubiquitinates p53 in vitro. Immunoblot of ubiquitinated p53 incubated with or without USP28 in an in vitro deubiquitination assay (see Materials and methods). Note the reduction in the polyubiquitinated form of p53 in the presence of USP28. (**E**) High levels of USP28 can ectopically stabilize nuclear p53 in the absence of mitotic stress. Immunofluorescence images of cells stained with the indicated antibodies. Expression of wild type USP28 or USP28$^{CI}$ was induced in *PLK4$^{as}$; USP28$^{-/-}$* cells with 10 ng/μl (low level expression) or 100 ng/μl (overexpression) of doxycycline for two days before cell fixing and staining. The percentage in the merged panel indicates the proportion of cells with p53 nuclear accumulation. Scale bar, 5 μm. (**F**) Nuclear p53 accumulation caused by USP28$^{WT}$ overexpression is not due to increased p53 transcription. Quantification of p53 mRNA levels relative to GAPDH in (**E**) by qRT-PCR. Data are means ± SD. *n* = 6 from two independent experiments. (**G**) USP28 ectopically stabilizes nuclear p53 independently of 53BP1. Immunofluorescence images of cells stained with the indicated antibodies. Wild type USP28 was induced in *PLK4$^{as}$; 53BP1$^{-/-}$* cells with 10 ng/μl (low level expression) or 100 ng/μl (overexpression) of doxycycline for two days. The percentage in the merged panel indicates the proportion of cells with p53 nuclear accumulation. Scale bar, 5 μm. (**H**) Overexpression of 53BP1 can ectopically stabilize nuclear p53 in the absence of mitotic stress.

*Figure 4 continued on next page*

*Figure 4 continued*

Immunofluorescence images of cells stained with the indicated antibodies. Expression of wild type 53BP1 or 53BP1$^{ΔBRCT}$ was induced in *PLK4$^{as}$*; *53BP1$^{-/-}$* cells with 10 ng/μl (low level expression) or 1 μg/μl (overexpression) of doxycycline for two days before cell fixing and staining. The percentage in the merged panel indicates the proportion of cells with p53 nuclear accumulation. Scale bar, 5 μm. (I) Ectopic stabilization of nuclear p53 by 53BP1 requires USP28. Immunofluorescence images of cells stained with the indicated antibodies. Wild type 53BP1 was induced in *PLK4$^{as}$*; *USP28$^{-/-}$* cells with 10 ng/μl (low level expression) or 1 μg/μl (overexpression) of doxycycline for two days. The percentage in the merged panel indicates the proportion of cells with p53 nuclear accumulation. Scale bar, 5 μm.
The following figure supplements are available for figure 4:

**Figure supplement 1.** USP28$^{WT}$ and USP28$^{CI}$ are exogenously expressed to similar levels in *PLK4$^{as}$*; *USP28$^{-/-}$* cells.
**Figure supplement 2.** Schematic outlining the timeline of drug treatments used.

reactivation of SAC efficiently targeted both BubR1 and Mad2 back to the kinetochore, but had no effect on 53BP1 (*Figure 5C*), further demonstrating that unlike SAC, the disassociation of 53BP1 from kinetochore is irreversible, a distinct property that can potentially be used to mark the duration of mitotic stress independent of SAC.

We next examined the localization of 53BP1 or USP28 in interphase cells that have gone through either stressed or normal mitosis. To avoid cell cycle interference, we examined the localization of 53BP1 and USP28 in cycling *p21$^{-/-}$* cells with or without centrosomes. In this setup, cells in both conditions are actively proliferating, but with one experiencing mitotic stress and p53 activation, and the other not. Strikingly, unlike unstressed *p21$^{-/-}$* cells that mostly lacked nuclear p53, in the stressed condition, p53 not only was stabilized in the nucleus, but also formed bright nuclear foci of various sizes co-localizing with 53BP1 and USP28 in ~30% of the cell population (*Figure 5D*), suggesting that 53BP1, USP28 and p53 interact with each other after a stressed mitosis, consistent with the known interaction between 53BP1 and p53 or USP28 (*Thukral et al., 1994*; *Iwabuchi et al., 1994*; *Zhang et al., 2006*).

## 53BP1/USP28 acts independent of SAC, and vice versa

53BP1/USP28 selects against stressed or delayed mitosis while SAC ensures correct mitosis by delaying mitotic progression. The contrary nature of the two processes suggests that 53BP1/USP28 and SAC may act independent of each other. Indeed, SAC-dependent mitotic delay caused by centrosome loss or Eg5 inhibition can still occur normally in *p53$^{-/-}$*, *53BP1$^{-/-}$*, *USP28$^{-/-}$* and *p21$^{-/-}$* cells (*Figures 1G*, *2C*, *5A* right and *Figure 5—figure supplement 1B*; not shown). Similarly, we found that the proteasome inhibitor MG132 could arrest cells at metaphase without activating SAC, as indicated by the weak BubR1 or lack of Mad2 signals at the kinetochores (*Figure 5C*), but efficiently trigger G1 arrest as shown above (*Figure 2H*, bottom panels), consistent with a previous report (*Uetake and Sluder, 2010*). To further test if a transient SAC activation during early prometaphase is required for the G1 arrest, we repeated the MG132-induced mitotic delay assay in the presence or absence of MPS1 activity, a kinase essential for SAC activation (*Maciejowski et al., 2010*) (*Figure 5—figure supplement 2C*). Using *MPS1$^{as}$* cells, we found that inactivation of MPS1, which disabled SAC (*Figure 5E*), caused all cells to rapidly progress through mitosis in 12 min (*Maciejowski et al., 2010*) (not shown), and became fully arrested in the following G1 (*Figure 5F*, top panel, 0 hr), likely due to erroneous chromosome segregation or cell division which resulted from the drastically shortened mitosis (*Maciejowski et al., 2010*). Consistent with this notion, we found that such MPS1 deficiency-induced G1 arrest could be reversed by a short treatment of MG132 in mitosis for 0.5 hr (*Figure 5F*, top panel), allowing cells sufficient time to assemble spindles for cell division. Intriguingly, however, longer MG132 treatments of 2 hr or more, which allowed cells more than enough time for spindle assembly, uniformly arrested cells in G1 after mitosis, regardless of whether MPS1 activity was present or not (*Figure 5F*, top panel). Consistently, removal of 53BP1 rescued the G1 arrest induced by the prolonged MG132 treatment in the absence of MPS1 activity (*Figure 5F*, bottom panel). These results together suggest that a prolonged mitotic stress can activate G1 arrest without ever activating SAC. Thus, 53BP1/USP28 and SAC are two in-parallel mitotic

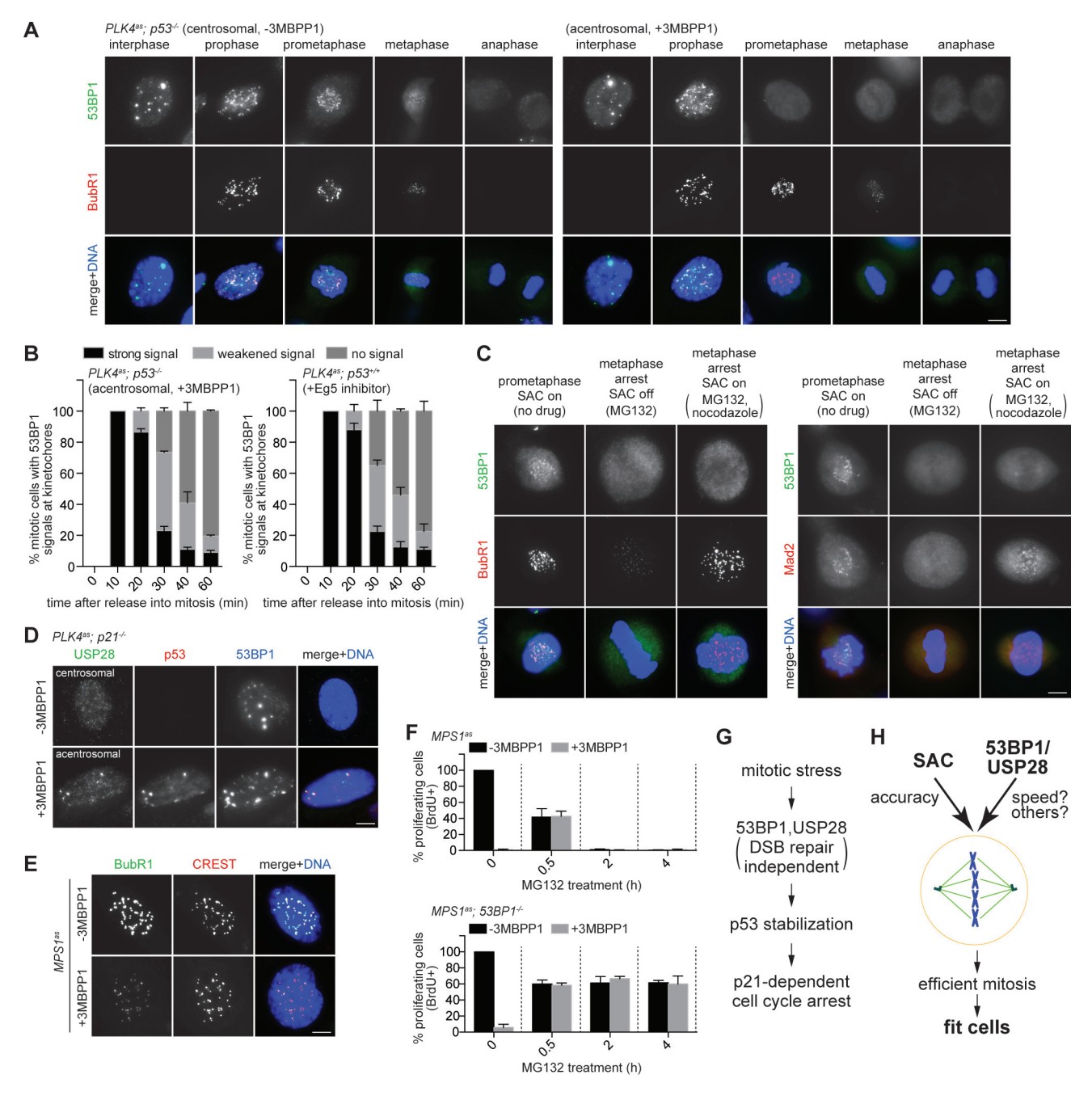

**Figure 5.** SAC activation is not essential for mitotic stress-induced G1 arrest. (**A**) 53BP1 normally disassociates from the kinetochores during anaphase, but can do so prematurely in prometaphase upon mitotic stress/delay. Immunofluorescence images of centrosomal (left) or acentrosomal (right) cells going through mitosis stained with the indicated antibodies. Bright BubR1 signals in prophase and prometaphase indicate SAC activation. Scale bar, 5 μm. (**B**) Disassociation of 53BP1 from mitotic kinetochores in cells experiencing mitotic stress/delay is time dependent. Quantification of proportion of mitotic cells with 53BP1 localization at the kinetochores. Cells arrested in G2/M were released into mitosis without centrosome (left), or with the spindle poison Eg5 inhibitor (right) as indicated, and then harvested at various time points after the release. Cells were stained with DAPI and antibodies against 53BP1 and BubR1 for scoring. Data are means ± SD. *n*>130, *N* = 3. (**C**) Disassociation of 53BP1 from kinetochores during mitotic stress is not reversible. Cells arrested in metaphase with MG132 for 4 hr followed by reactivation of SAC with nocodazole treatment for 10 min were stained with the indicated antibodies. Bright BubR1 or Mad2 signals indicate SAC activation. Scale bar, 5 μm. (**D**) 53BP1, USP28 and p53 form large nuclear foci in response to mitotic delay. *PLK4^{as}; p21^{-/-}* cells proliferating with or without 3MBPP1 stained with the indicated antibodies. Scale bar, 5 μm. (**E**) SAC is inactive in *MPS1^{as}* cells treated with 3MBPP1. *MPS1^{as}* cells treated with 3MBPP1 and Cdk1 inhibitor (RO-3306) for 18 hr were released into mitosis and were processed for immunofluorescence to visualize BubR1 and CREST. Note that SAC activation was disabled by MPS1 inhibition (weak BubR1

*Figure 5 continued on next page*

*Figure 5 continued*

signals). Scale bar, 5 μm. (**F**) MG132 treated cells arrest in G1 in the absence of SAC activity. BrdU incorporation assay showing proportion of proliferating *MPS1^as^* or *MPS1^as^; 53BP1^-/-^* cells, with or without 3MBPP1, following release into mitosis with different length of MG132 treatment. Data are means ± SD. Percentages are normalized to the untreated control. *n*>250, *N* = 3. (**G**) Model of 53BP1 and USP28 transducing mitotic stresses into p53 stabilization and p21 dependent cell cycle arrest. (**H**) A model proposing that the independent collaboration of SAC and 53BP1/USP28 drives efficient mitosis and cell fitness.

The following figure supplements are available for figure 5:

**Figure supplement 1.** 53BP1 delocalizes from kinetochores in prometaphase in response to mitotic stress.

**Figure supplement 2.** Schematic outlining the timeline of synchronization and drug treatments used.

programs, with SAC protecting the accuracy of mitosis (at the expense of speed), and 53BP1/USP28 selecting against stressed/delayed mitosis (see *Figure 5H*).

## Discussion

Mitosis is a crucial phase of the cell cycle in which cellular elements are divided between two daughter cells. While the accuracy of chromosome segregation is guarded by SAC, we found that a separate program involving 53BP1 and USP28 is required to monitor mitotic progression independent of SAC. We showed that 53BP1 and USP28 are required to trigger p53/p21- dependent cell cycle arrest, evoking an irreversible stress response that selects against unfit cells with disturbed mitosis (*Figure 5G*). Interestingly, 53BP1 is a well-characterized DDR component whose activity in DNA repair is disabled specifically during mitosis (*Orthwein et al., 2014*). Consistently, we found that the involvement of 53BP1 in G1 arrest induced by disturbed mitosis is independent of its role in DNA repair, but requires its known binding partners p53 and USP28, unraveling the functional significance of these previously puzzling interactions. Thus, 53BP1 is more than a DNA damage-responding molecule. Moreover, we showed that the kinetochore association of 53BP1 during mitosis is time-sensitive, providing a possible route through which the duration of mitotic stress can be measured or responded to, directly or indirectly, in the form of chemical reactions. It remains to be seen if delocalization of 53BP1 from kinetochores directly activates the G1 arrest induced by mitotic stress. Note that while centrosome loss frequently lengthens the duration of mitotic progression, previous studies showed that G1 arrest induced by centrosome loss can occur without a significant mitotic delay (*Lambrus et al., 2015*; *Wong, 2015*). It is therefore possible that both centrosome loss and mitotic delay (induced by Eg5 inhibitor or MG132) cause a common underlying damage that activates the G1 arrest. The underlying damage is yet to be identified. Alternatively, two distinct errors may be generated by centrosome loss and prolonged mitosis, and the G1 arrest induced by both errors are mediated by 53BP1 and USP28.

The accuracy of mitosis is guarded by SAC at the expense of speed. That is, when mitosis is disturbed, SAC is activated to stall the process, allowing more time for mitosis to proceed correctly. In this sense, cells should not be 'penalized' for spending extra time in mitosis, when accuracy can evidently benefit from it. This idea, however, is inconsistent with the observation that even a mild delay in mitosis (~90 min) can uniformly trigger p53 dependent cell cycle arrest. One possibility is that mitosis is a delicate environment where non-specific cellular damages are prone to be induced (*Hayashi et al., 2012*; *Orth et al., 2012*; *Colin et al., 2015*), and that p53 activation may act as a 'preemptive strike' to eliminate cells potentially damaged by spending too much time in mitosis. Another non-mutually exclusive possibility is that slow mitosis may serve as an indication for cells that are unfit for proliferation (i.e. overstressed, aged, or carrying unresolved damages) but somehow slip into mitosis, and thus need to be removed from the population. In this case, mitosis may be used as a fitness test for cycling cells, providing a selection where only cells able to exercise mitosis efficiently are allowed to proliferate again. It will be very interesting to see how cells properly manage this tug-of-war between accuracy and speed to maximize mitotic efficiency (*Figure 5H*), and what will happen if the balance breaks off. Our results here will likely facilitate future studies on subjects related to mitotic efficiency, stress response, cell fitness, and antimitotic drug development.

# Materials and methods

## Cell culture

RPE1 cells were cultured in DME/F-12 (1:1) medium supplemented with 10% FBS and 1% penicillin-streptomycin. The RPE1 tetracycline-inducible $PLK4^{as}$ cells ($PLK4^{-/-}$; $tet\text{-}PLK4^{as}$) were grown under a constant supply of 5 ng/ml of doxycycline replaced every two days to support centrosome biogenesis. We also obtained $PLK4^{as}$ *knock-in* RPE1 cells (see below) in which centrosomes biogenesis are supported by $PLK4^{as}$ expressed under the endogenous promoter. The $PLK4^{-/-}$; $tet\text{-}PLK4^{as}$ cell line was used in our eight independent CRISPR-mediated screens, and $PLK4^{as}\text{-}KI$ cells were used to validate our screen in other assays. To inhibit $PLK4^{as}$, 2 µM of 3MBPP1 was added to the media. Cells were arrested in mitosis with Eg5 inhibitor III, Dimethylenastron, or MG132 at 1 µM or 10 µM, respectively. Microtubule depolymerization was achieved through 200 ng/ml nocodazole treatment. BrdU was added to the culture medium at 30 µM for 24 hr before fixation to label proliferating cells. Cdk1 inhibitor RO-3306 was used at 10 µM to arrest cells at G2/M boundary.

## Cell lines and plasmid constructs

The RPE1 tetracycline-inducible $PLK4^{as}$ cell line ($PLK4^{-/-}$; $tet\text{-}PLK4^{as}$) was generated in our lab (see below for details) (*Kim, 2016*). The RPE1 cell line carrying a knock-in $PLK4^{as}$ allele ($PLK4^{as}\text{-}KI$) was a kind gift from Andrew J. Holland (*Moyer et al., 2015*). $PLK4^{-/-}$; $tet\text{-}PLK4^{as}$ was used in the genome-wide CRISPR screen and the experiments in *Figure 1*, whereas $PLK4^{as}\text{-}KI$ was used for the experiments in *Figures 2–5*. We obtained the RPE1 $MPS1^{as}$ cells from Prasad V. Jallepalli (*Maciejowski et al., 2010*). RPE1 is not a commonly misidentified cell line and we have not authenticated the RPE1 cell lines we used. There is no mycoplasma contamination in the RPE1 cell lines. Stable p53$^{-/-}$, 53BP1$^{-/-}$, USP28$^{-/-}$, and p21$^{-/-}$ knockout cell lines derived from both $PLK4^{-/-}$; $tet\text{-}PLK4^{as}$ and $PLK4^{as}\text{-}KI$ cells were made by CRISPR (see below) for validation and rescue experiments, and results from $PLK4^{as}\text{-}KI$ derived cells were shown. For rescue experiments, clonal $PLK4^{as}\text{-}KI$; 53BP1$^{-/-}$ or $PLK4^{as}\text{-}KI$; USP28$^{-/-}$ cell lines stably carrying various constructs expressing 53BP1$^{WT}$, 53BP1$^{\Delta BRCT}$, 53BP1$^{TESE}$, 53BP1$^{TASA}$, USP28$^{WT}$ or USP28$^{CI}$ from the tetracycline-inducible promoter were made through in vivo gene delivery using the lentiviral vector pLVX-Tight-Puro vector (Clonetech). 53BP1$^{WT}$, 53BP1$^{\Delta BRCT}$, 53BP1$^{TESE}$ and 53BP1$^{TASA}$ cDNA constructs were kind gifts from Daniel Durocher (*Orthwein et al., 2014*), and were used for subcloning into pLVX-Tight-Puro vector. Wild type USP28 construct (pDZ50, Addgene plasmid #41948) was a gift from Stephen Elledge (*Zhang et al., 2006*), and was used for subcloning into pLVX-Tight-Puro vector. USP28$^{CI}$ was created with site-directed mutagenesis (Stratagene).

## Tetracycline-inducible $PLK4^{as}$ cell line generation

Both *PLK4* loci in RPE1 cells were modified through homologous recombination using adeno-associated virus vectors as described previously (*Tsou et al., 2009*). A region covering exon 3 and 4 of *PLK4* loci, which contains the catalytic site of the kinase domain, was flanked with LoxP (or floxed). After generating the $PLK4^{flox/neoflox}$ clones, we utilized the lentiviral pLVX-Tight-Puro vector system (Clonetech) to transduce a tetracycline inducible, analog sensitive construct of PLK4 ($PLK4^{as}$). These $PLK4^{flox/neoflox}$; $tet\text{-}PLK4^{as}$ cells were then infected with an adenovirus expressing Cre recombinase to delete the endogenous *PLK4*, plated in 96 well plates, and cultured in media containing 5 ng/ml of doxycycline. Clonal $PLK4^{-/-}$; $tet\text{-}PLK4^{as}$ cell lines that exhibited the normal number of centrosomes were selected and maintained under 5 ng/ml of doxycycline.

## Genome-wide CRISPR-mediated loss-of-function screen

The human Genome-scale CRISPR Knock-Out (GeCKO) v2.0 pooled libraries generated by Zhang Lab were acquired from Addgene (#1000000049, 2 vector system - lentiCas9-Blast and lentiGuide-Puro) (http://www.addgene.org/crispr/libraries/geckov2/) (*Sanjana et al., 2014*; *Shalem et al., 2014*). Amplification of the libraries was performed as recommended using MegaX DH10B T1 Electrocomp Cells (Invitrogen). Lentivirus library was produced using calcium-phosphate transfection in HEK293T cells. For each CRISPR screen, five 10 cm plates were seeded with the tetracycline-inducible $PLK4^{as}$ cells ($PLK4^{-/-}$; $tet\text{-}PLK4^{as}$) expressing Cas9 at $1 \times 10^6$ cells/plate. Cells were transduced with the lentivirus library the next day at 500 µl/plate for 5 hr. The five 10 cm plates with the

transduced cells were split into thirty 15 cm plates the day after. Three days later, to induce centrosome loss, doxycycline was removed to turn off PLK4[as] expression and 3MBPP1 was added to inactivate any basal level PLK4[as] activity. Cells were cultured and propagated for an additional four weeks before genomic DNA was extracted for deep sequencing to identify enriched sgRNAs. Cells were also harvested two days after PLK4 inactivation to monitor baseline sgRNA distribution. Eight independent CRISPR screens were performed to facilitate discrimination between true and false positive hits. Because off-target effects and passenger mutations can cause spuriously high reads for some sgRNAs in each experiment, a single high count for one sgRNA against a gene in one experiment should not be regarded as a true hit. For a gene to be scored as a 'positive' hit, at least two of its six sgRNAs needed to be repeatedly enriched (greater than 1000 HiSeq reads) in independent screens, with the HiSeq reads at least three times higher than the average of the baseline read.

## CRISPR-mediated gene targeting

RNA-guided targeting of *p53, 53BP1, USP28*, and *p21* in human cells was achieved through coexpression of the Cas9 protein with gRNAs using reagents prepared by the Church group (*Mali, 2013*), which are available from Addgene (http://www.addgene.org/crispr/church/). Sequences of gRNAs used are as follows: p53 (5'-GGGCAGCTACGGTTTCCGTCTGG-3'), 53BP1 (5'-GTATACCTGCTTG TCCTGTT-3', 5'-CTGCTCAATGACCTGACTGA-3'), USP28 (5'-ATCAACTCTCCTCCAGTCAT-3', 5'-TGAGCGTTTAGTTTCTGCAG-3'), and p21 (5'-CCATTAGCGCATCACAGTCG-3', 5'-AGTCGAAG TTCCATCGCTCA-3'). All gRNAs were cloned into the gRNA Cloning Vector (Addgene plasmid #41824) via the Gibson assembly method (New England Biolabs) as described previously (*Mali, 2013*). 5 µg Cas9 plasmid (Addgene plasmid #41815) and 5 µg gRNA were nucleofected according to manufacturer's instructions (Lonza, Basel, Switzerland). Cells were examined for the loss of proteins 7 days after nucleofection.

## CRISPR cell line genotyping

Primers were designed for PCR amplification of genomic DNA containing each sgRNA target site: 53BP1 (5'-ACAGCTGGAGAAGAACGAGG-3', 5'-CCTCCCAGGTTCAAGCAACT-3'), USP28 (5'-TGGGCAATTTGGAGGCTCTT-3', 5'-TGTCGCCTACCTGGATAGCT-3') and p21 (5'-CCAGGGC TGCGATTAGGAAA-3', 5'-GCAAAGGGCCTGGCATAATG-3'). PCR products of ~800–1100 bp were cloned into PCR-TOPO-TA cloning vector (Invitrogen). Plasmids were then sequenced and indels identified through sequence alignment.

## Antibodies

Antibodies used in this study were listed with the information on working dilution and source in parentheses – anti-p53 (rabbit, 1:500, sc-6243, Santa Cruz Biotechnology; mouse, 1:200, sc-2025, Santa Cruz Biotechnology), anti-p21 (rabbit, 1:200, ab7960, Abcam), anti-actin (rabbit, 1:10000, A2066, Sigma), anti-53BP1 (rabbit, 1:1000, NB100-304, Novus Biologicals; mouse 1:1000, MAB3802, Millipore), anti-USP28 (rabbit, 1:200, A300-898A, Bethyl Laboratories), anti-BrdU (rat, 1:500, MCA2060T, AbD Serotec), anti-BubR1 (mouse, 1:200, ab4637, Abcam), anti-Mad2 (rabbit, 1:500, A300-301A, Bethyl Laboratories), anti-CREST (human, 1:1000, HCT-0100, ImmunoVision), anti-HA (mouse, 1:1000, MMSH101P, Covance), anti-GFP FITC conjugated (goat, 1:1000, 600-102-215, Rockland), anti-centrin2 (mouse, 1:1000, 04–1624, Millipore), anti-γ-tub (mouse, 1:500, sc-51715, Santa Cruz Biotechology) and anti-γ-H2AX (mouse, 1:200, 05–636, Millipore). Secondary antibodies Alexa-Fluor 488, 594, and 680 were from Molecular Probes.

## Immunofluorescence and microscopy

Cells were washed once in phosphate-buffered saline (PBS) then fixed in ice-cold methanol at −20°C for 10 min. Slides were blocked with 3% bovine serum albumin (w/v) with 0.1% Triton X-100 in PBS before incubating with primary antibodies. For BrdU staining, cells were treated with 2N HCl at room temperature for 30 min followed by rinsing in PBS before anti-BrdU incubation. DNA was visualized using 4',6-diamidino-2-phenylindole (DAPI). Fluorescent images were acquired on an upright microscope (Axio imager; Carl Zeiss) equipped with 100x oil objectives, NA of 1.4, and a camera (ORCA ER; Hamamatsu Photonics). Captured images were processed with Axiovision (Carl Zeiss) and Photoshop CS5 (Adobe).

## Mitotic duration assay

Cells were plated in 6-wells plates with or without 3MBPP1 the day prior to live-cell imaging. Images were acquired on an inverted miscroscope (Axiovert; Carl Zeiss) equipped with a 10x phase objective, motorized temperature-controlled stage, environmental chamber, $CO_2$ enrichment system (Carl Zeiss), and a camera (ORCA ER; Hamamatsu Photonics). Image acquisition and processing were performed using Axiovision software (Carl Zeiss). Images were acquired every 7 min for a duration of 24 hr. Mitotic duration was defined as the period between cell rounding and cell division.

## Growth curve analysis

Cells were seeded at 500,000 cells per 10 cm plates at day 0. 3MBPP1 was added accordingly. Cells were trypsinized for cell counting using a haemocytometer every two days for eight days, and reseeded with appropriate dilution for subsequent counting.

## Determination of p53 protein levels

Acentrosomal cells were harvested seven days after 3MBPP1 treatment. All cells were washed once in PBS and directly lysed on plates with SDS loading buffer (125 mM Tris-HCl pH 6.8, 50% (v/v) glycerol, 4% (w/v) SDS, 0.02% (w/v) bromophenol blue, 100 mM DTT). Genomic DNA was sheared by passing through a 27G needle. Lysates were resolved by SDS-PAGE.

## p53 in vitro deubiquitination assay

Recombinant USP28 protein was purchased from BostonBiochem. To perform the deubiquitination assay, USP28 protein (2 µM) was incubated in the reaction buffer containing 10 mM DTT for 30 min followed by addition of ubiquitinated p53 as the substrate (100 nM) produced with the MDM2 Ubiquitin Ligase Kit (BostonBiochem, MA; K200b). Reaction was done at 37°C for 90 min. Reaction was terminated by addition of 5X loading buffer (SDS-PAGE sample buffer) and 1M DTT, and heated for 5 min at 90°C. Ubiquitinated-p53 was detected using standard western blot technique with anti-p53 antibodies.

## qRT-PCR

Total RNA was isolated using RNeasy Plus Mini Kit (Qiagen). cDNA was generated using qScript cDNA SuperMix (Quanta Biosciences). Real-time PCR was carried out using PerfeCTa SYBR Green SuperMix (Quanta Biosciences) on ViiA 7 Real-Time PCR System (Applied Biosystems). GAPDH was used as an endogenous normalization control. Primers used were p53 (5′-AGAGTCTATAGGCC-CACCCC-3′, 5′-GCTCGACGCTAGGATCTGAC-3′) and GAPDH (5′-GCGAGATCCCTCCAAAATCAA-3′, 5′-GTTCACACCCATGACGAACAT-3′).

# Acknowledgement

We thank A Holland for the knock-in *PLK4[as]* cell line, A Orthwein and D Durocher for 53BP1 constructs, P Jallepalli for the RPE1 *MPS1[as]* cell line, J Li and A Viale at Memorial Sloan Kettering Cancer Center Integrated Genomics Operation for assistance with Hi-Seq, and C Haynes and E Foley for comments on the manuscript. This work was supported by the National Institute of Health grant GM088253 and American Cancer Society grant RSG-14-153-01-CCG to MFBT. MFBT was also supported by Geoffrey Beene Cancer Research Center.

# Additional information

### Funding

| Funder | Grant reference number | Author |
| --- | --- | --- |
| Geoffrey Beene Cancer Research Center | | Tuhin Das |
| National Institutes of Health | GM088253 | Meng-Fu Bryan Tsou |
| American Cancer Society | RSG-14-153-01-CCG | Meng-Fu Bryan Tsou |

The funders had no role in study design, data collection and interpretation, or the decision to submit the work for publication.

## Author contributions

CSF, Conception and design, Acquisition of data, Analysis and interpretation of data, Drafting or revising the article; GM, TD, JG, Conception and design, Acquisition of data, Analysis and interpretation of data; MK, BPO, DI, Acquisition of data, Analysis and interpretation of data, Contributed unpublished essential data or reagents; M-FBT, Conception and design, Analysis and interpretation of data, Drafting or revising the article

## Author ORCIDs

Meng-Fu Bryan Tsou, http://orcid.org/0000-0002-2159-8836

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
