## [Decision Letter]

Thank you for submitting your article "53BP1 and USP28 mediate p53 dependent cell cycle arrest in response to mitotic stress" for consideration by *eLife*. Your article has been favorably evaluated by Tony Hunter (Senior editor) and three reviewers, one of whom, Jon Pines (Reviewer #1), is a member of our Board of Reviewing Editors.

The reviewers have discussed the reviews with one another and the Reviewing Editor has drafted this decision to help you prepare a revised submission.

Summary:

In this study the authors have investigated the response of cells to mitotic stress. They treat an RPE1 *Plk4^as^* cell line with analogue inhibitor to deplete centrosomes and find that cells arrest in G1 with high p53 and p21 levels. This arrest is p53 dependent as previously described (e.g. Holland, Oegema), and in a very nicely done CRISPR screen the authors identify 27 clones that by-pass p53 to keep proliferating. Of these genes the authors follow up 53BP1 and USP28 in this study. They show that both proteins are required for p53 to accumulate in cells, and for cells to arrest in G1 cells after a delay in mitosis. They further show that 53BP1 lacking its C-terminal BRCT domains cannot arrest, whereas a 53BP1 T1609E/S1618E phosphomimetic mutant can impose a G1 arrest but is unable to carry out the M-phase DDR. USP28 requires its DUB catalytic activity for the G1 arrest, and this correlates with the ability to stabilize p53. Over-expressing USP28 can stabilize p53 in the absence of 53BP1, but 53BP1 needs USP28. These data are clear and nicely support the authors' conclusions that 53BP1 and USP28 act upstream of p53 to stabilize it in response to 'mitotic stress'.

Essential revisions:

1) The major problem with the paper is how it is currently written. The Abstract and Introduction are very limited and oversimplified, especially in conceptual terms, as such, following the narrative through the rest of the manuscript is incredibly difficult and the readers of *eLife* will be very confused if this is not addressed with a major re-write. In particular, the authors invoke the term 'Mitotic Stress Response' and in doing so they conflate a series of very separate phenomena. There are multiple mechanisms that operate before, during and after mitosis in response to various "stresses". In chronological order these are the "antephase" checkpoint (e.g. Rieder, Pines); the Spindle Assembly Checkpoint, which delays anaphase until all kinetochores are stably attached to spindle microtubules, and many "stresses" that delay mitosis do so by activating the SAC; apoptosis that can occur in mitosis if the "stress" that is activating the SAC cannot be corrected (Taylor, Malumbres); finally the post-mitotic response, which may reflect one or more molecular pathways depending on the nature of the stress, and can induce either apoptosis or a cell cycle block. After mitosis, if a cell missegregates a chromosome, a chromosome trapped in the cleavage furrow can activate an ATM-dependent DNA damage response in the next G1 (e.g. Medema). If a missegregating chromosome ends up in a micronucleus, this can induce a ATR-dependent DNA damage response when the cell goes into the next S phase (e.g. Pellman). Missegregation of a whole chromosome can induce a proteotoxic response in turn influencing cell cycle progression (e.g. Amon, Compton). Mitotic errors that cause cytokinesis failure can also result in p53 activation via the Hippo pathway (e.g. Pellman).

In addition, to all these pathways, if a cell simply spends too long in mitosis, this will influence the subsequent G1 (pioneering studies by Greenfield Sluder) and the authors suggest that this mechanism is at play here. But they do not directly show that cells arrest because of the time that they spend in mitosis and that this is by-passed when 53BP1 or USP28 is mutated. Note also that studies from both the Holland and Oegema labs conclude that arrest after centriole loss is not a result of prolonged mitosis. The authors should rewrite their manuscript to clarify the phenomenon they are studying and put it into proper context.

2) The nature of the stress signal detected is unclear. The authors suggest that 53BP1 may be acting as a mitotic timer based on immunofluorescence experiments in which they assay 53BP1 localization to kinetochores. This localization is not directly related to the attachment state of the kinetochores but what regulates the localization is not clear. The problem is that the experiments are based on release from a long block with RO-3306, which could perturb centromere behavior and kinetochore assembly, and the assay is based on a qualitative assessment of strong versus weak signals. Moreover, the choice of cell line for these experiments is not ideal; in Figure 5, localization of 53BP1 is done in p53-deficient cells. In Figure 5, one graph is done with wild type while the other is done with mutant. Note that there is no direct evidence to link 53BP1 de-localization from the kinetochore to p53 activation in the next G1 phase. To support their contention that 53BP1 acts as a timer the authors will need to assay its localization by time-lapse microscopy. Alternatively, the authors should tone down their conclusions and mention the caveats to these experiments.

3) p53 mRNA levels need to be measured to control for the steady state protein levels in Figure 2 and Figure 4 to strengthen the conclusion that USP28 is directly regulating p53 ubiquitylation.

[Editors' note: further revisions were requested prior to acceptance, as described below.]

Thank you for resubmitting your work entitled "53BP1 and USP28 mediate p53 dependent cell cycle arrest in response to centrosome loss and prolonged mitosis" for further consideration at *eLife*. Your revised article has been favorably evaluated by Tony Hunter as the Senior editor and a Reviewing editor.

The manuscript has been improved but there is one remaining issue that need to be addressed before acceptance. As outlined in the previous decision letter, the Introduction fails to review adequately what is known about the response of dividing cells to stresses. The revised Introduction is again too brief and fails to do justice to previous work in this area. There are also a number of typographic and grammatical mistakes indicating that it may have been written in a rush. We realise that the authors are concerned about competing papers but a rapid review process should not be at the expense of scholarship.

---

## [Author Response]

Essential revisions:

*1) The major problem with the paper is how it is currently written. The Abstract and Introduction are very limited and oversimplified, especially in conceptual terms, as such, following the narrative through the rest of the manuscript is incredibly difficult and the readers of eLife will be very confused if this is not addressed with a major re-write. In particular, the authors invoke the term 'Mitotic Stress Response' and in doing so they conflate a series of very separate phenomena. There are multiple mechanisms that operate before, during and after mitosis in response to various "stresses". In chronological order these are the "antephase" checkpoint (e.g. Rieder, Pines); the Spindle Assembly Checkpoint, which delays anaphase until all kinetochores are stably attached to spindle microtubules, and many "stresses" that delay mitosis do so by activating the SAC; apoptosis that can occur in mitosis if the "stress" that is activating the SAC cannot be corrected (Taylor, Malumbres); finally the post-mitotic response, which may reflect one or more molecular pathways depending on the nature of the stress, and can induce either apoptosis or a cell cycle block. After mitosis, if a cell missegregates a chromosome, a chromosome trapped in the cleavage furrow can activate an ATM-dependent DNA damage response in the next G1 (e.g. Medema). If a missegregating chromosome ends up in a micronucleus, this can induce a ATR-dependent DNA damage response when the cell goes into the next S phase (e.g. Pellman). Missegregation of a whole chromosome can induce a proteotoxic response in turn influencing cell cycle progression (e.g. Amon, Compton). Mitotic errors that cause cytokinesis failure can also result in p53 activation via the Hippo pathway (e.g. Pellman).*

In addition, to all these pathways, if a cell simply spends too long in mitosis, this will influence the subsequent G1 (pioneering studies by Greenfield Sluder) and the authors suggest that this mechanism is at play here. But they do not directly show that cells arrest because of the time that they spend in mitosis and that this is by-passed when 53BP1 or USP28 is mutated. Note also that studies from both the Holland and Oegema labs conclude that arrest after centriole loss is not a result of prolonged mitosis. The authors should rewrite their manuscript to clarify the phenomenon they are studying and put it into proper context.

We agree that the term ‘mitotic stress response’ could be ambiguous and confusing to readers and have removed it from the manuscript. We have now rewritten the manuscript to describe the events we are studying as centrosome loss- or prolonged mitosis-induced G1 arrest.

We have also included words in the Discussion about the relationship between centrosome loss, mitotic delay, and cell cycle arrest: “Note that while centrosome loss frequently lengthens the duration of mitotic progression, previous studies showed that G1 arrest induced by centrosome loss can occur without a significant mitotic delay [Labrus et al., 2015; Wong et al., 2015]. It is therefore possible that both centrosome loss and mitotic delay (induced by Eg5 inhibitor or MG132) cause a common underlying damage that activates the G1 arrest. The underlying damage is yet to be identified. Alternatively, two distinct errors may be generated by centrosome loss and prolonged mitosis, and the G1 arrest induced by both errors are mediated by 53BP1 and USP28.”

2) The nature of the stress signal detected is unclear. The authors suggest that 53BP1 may be acting as a mitotic timer based on immunofluorescence experiments in which they assay 53BP1 localization to kinetochores. This localization is not directly related to the attachment state of the kinetochores but what regulates the localization is not clear. The problem is that the experiments are based on release from a long block with RO-3306, which could perturb centromere behavior and kinetochore assembly, and the assay is based on a qualitative assessment of strong versus weak signals. Moreover, the choice of cell line for these experiments is not ideal; in Figure 5, localization of 53BP1 is done in p53-deficient cells. In Figure 5, one graph is done with wild type while the other is done with mutant. Note that there is no direct evidence to link 53BP1 de-localization from the kinetochore to p53 activation in the next G1 phase. To support their contention that 53BP1 acts as a timer the authors will need to assay its localization by time-lapse microscopy. Alternatively, the authors should tone down their conclusions and mention the caveats to these experiments.

We agree with reviewers’ concern about the lack of direct evidence in our conclusion that 53BP1 delocalization from kinetochores leads to p53 activation in G1. The statement of kinetochore 53BP1 as the ‘mitotic timer’ has been removed from the Abstract. We leave this intriguing observation for discussion.

3) p53 mRNA levels need to be measured to control for the steady state protein levels in Figure 2 and Figure 4 to strengthen the conclusion that USP28 is directly regulating p53 ubiquitylation.

We have examined p53 mRNA levels by qRT-PCR for the experiments in Figure 2 in the original submission) and 4E. We observe no increase in p53 mRNA levels during centrosome loss-induced G1 arrest (*PLK4^as^* + 3MBPP1), despite a marked elevation in p53 protein levels. This indicates that the accumulation of p53 protein is not due to upregulation of p53 gene transcription. Also of note is the general increase (1.4 to 2.5 fold) in p53 mRNA levels in *PLK4^as^; 53BP1^-/-^*and *PLK4^as^; USP28^-/-^*cells. This, however, does not culminate in an increase in p53 protein levels in the cells, suggesting that p53 protein levels are not tightly correlated with its gene transcription activity.

For experiment in 4E, we observe a modest increase (1.29 fold) in p53 mRNA levels in cells overexpressing USP28^WT^ (100 ng/ml doxycycline induction). A similar increase (1.25 fold) in p53 mRNA levels is also seen in cells overexpressing USP28^CI^. Yet, p53 nuclear accumulation is only detected in cells overexpressing USP28^WT^. This indicates that the increase in p53 mRNA levels is not the reason for the nuclear p53 stabilization seen in cells overexpressing USP28^WT^. Taken together, our qRT-PCR data support the conclusion that USP28 deubiquitinates p53 for protein stabilization. The results have been added to Figure 2 (panel G) and Figure 4 (panel F). The text and figure legends have been modified accordingly.

[Editors' note: further revisions were requested prior to acceptance, as described below.]

*The manuscript has been improved but there is one remaining issue that need to be addressed before acceptance. As outlined in the previous decision letter, the Introduction fails to review adequately what is known about the response of dividing cells to stresses. The revised Introduction is again too brief and fails to do justice to previous work in this area. There are also a number of typographic and grammatical mistakes indicating that it may have been written in a rush. We realise that the authors are concerned about competing papers but a rapid review process should not be at the expense of scholarship.*

We have rewritten and expanded the Introduction to include additional references as suggested by the reviewers. We have also made every effort to remove any typographical and grammatical mistakes in the manuscript.